

# Counter culture: causes, extent and solutions of systematic bias in the analysis of behavioural counts

Joel L. Pick[1,2,3], Nyil Khwaja[1,4], Michael A. Spence[1,5], Malika Ihle[1,6] and Shinichi Nakagawa[2]

[1] University of Sheffield, Sheffield, United Kingdom
[2] University of New South Wales, Sydney, Australia
[3] University of Edinburgh, Edinburgh, United Kingdom
[4] University of Canterbury, Christchurch, New Zealand
[5] Centre for Environmental Fisheries and Aquaculture Science, Lowestoft, United Kingdom
[6] Ludwig-Maximilians-Universität München, Munich, Germany

Corresponding author
Joel L. Pick, joel.l.pick@gmail.com

## ABSTRACT

We often quantify the rate at which a behaviour occurs by counting the number of times it occurs within a specific, short observation period. Measuring behaviour in such a way is typically unavoidable but induces error. This error acts to systematically reduce effect sizes, including metrics of particular interest to behavioural and evolutionary ecologists such as $R^2$, repeatability (intra-class correlation, ICC) and heritability. Through introducing a null model, the Poisson process, for modelling the frequency of behaviour, we give a mechanistic explanation of how this problem arises and demonstrate how it makes comparisons between studies and species problematic, because the magnitude of the error depends on how frequently the behaviour has been observed as well as how biologically variable the behaviour is. Importantly, the degree of error is predictable and so can be corrected for. Using the example of parental provisioning rate in birds, we assess the applicability of our null model for modelling the frequency of behaviour. We then survey recent literature and demonstrate that the error is rarely accounted for in current analyses. We highlight the problems that arise from this and provide solutions. We further discuss the biological implications of deviations from our null model, and highlight the new avenues of research that they may provide. Adopting our recommendations into analyses of behavioural counts will improve the accuracy of estimated effect sizes and allow meaningful comparisons to be made between studies.

## INTRODUCTION

Behaviour is frequently quantified by counting the number of times a specific event occurs within an observation period. This includes studies of parental care (*e.g.*, number of feeding visits), social interactions such as allopreening/grooming or aggression (*e.g.*, number of interactions) and mate choice (*e.g.*, number of copulations, courtship behaviours). Typically, when behaviour is quantified in such a way, we do not observe the total time

over which a behaviour takes place, and thus, all occurrences of the behaviour. Rather we sample a shorter time period in order to calculate a representative 'rate' at which the behaviour occurs. Take the example of parental provisioning behaviour in birds—although the nestling period may last 2 or more weeks, researchers typically record the feeding visits that occur in a shorter period of time, often in a 1 or 2 h period (*Murphy, Chutter & Redmond, 2015*). Measuring behaviour in this way is often unavoidable for practical reasons, nevertheless we accept that there will be some error in the quantification of a behaviour. Intuitively we also know that the longer we observe a behaviour, the better the representation of that behaviour we will have (*i.e.*, diminishing error with increasing sampling effort; see *Murphy, Chutter & Redmond, 2015*; *Lendvai et al., 2015*; *Sánchez-Tójar, Schroeder & Farine, 2018* for empirical evidence of this).

Evolutionary and behavioural ecologists are often interested in quantifying the total amount of variation that is due to differences between individuals and environments. We therefore frequently use metrics such as $R^2$ (proportion of total variance explained by a particular model), repeatability (the proportion of total variance due to individual identity effects; also known as the intraclass correlation coefficient - ICC) or heritability (proportion of total variance due to additive genetic effects) or, more broadly, quantify standardised effect sizes (*Nakagawa & Cuthill, 2007*; *Nakagawa & Schielzeth, 2010*; *Nakagawa & Schielzeth, 2013*). However, commonly used methods for analysing variation in behavioural counts fail to distinguish between the contribution of biological variation and the error introduced by sampling. Without taking this into account, we will both systematically *under* estimate effect sizes, and limit our ability to compare such metrics between studies, due to variation in sampling effort.

Here, we outline a broad model for thinking about how behavioural count data arise, which demonstrates the inevitability of systematic error in such data. We suggest a simple null model for describing these processes and discuss the problems associated with ignoring the stochasticity inherent in this data. Using the example of provisioning rate in birds, a frequently used, count-based measure of parental care, we show how behavioural counts fit our null model well. Through a literature survey we show how widespread the problem of not accounting for this stochastic error is. Finally we present some solutions to this problem. Despite the large focus of this article on provisioning, all the theoretical and practical issues and solutions described here are directly relevant to other behaviours sampled by counting events in a restricted time window.

## BEHAVIOUR AS A POINT PROCESS

Let us take a general example to describe our behavioural count data. Imagine that we want to describe the factors affecting the frequency (or rate) at which bees enter a nest. Here each arrival represents the behaviour of interest occurring. To do this, we watch the entrance to a nest for set observation periods (*e.g.*, 1 h). There will be many factors influencing the length of the intervals between arrivals (interval length), only a few of which we will realistically be able to quantify (*e.g.*, biotic factors such as the abundance and distribution of food/competitors/predators and abiotic factors such as

temperature/rainfall/wind/humidity at each moment during the this observation). As we are unable to describe the complex processes that lead to the timing of this behaviour, we can therefore describe the arrival of these bees (or more generally the occurrence of behaviour), as a stochastic process through time. Note that we are not suggesting that the behaviour arises through a stochastic process, rather that we can *describe* it as a stochastic process; although the behaviour may be completely deterministic, we do not have the information needed to describe it in this way. We therefore need a model that describes the stochastic distribution of events through time. This is broadly known as a point process, a probabilistic model for describing points (or events) in some space, for example the distribution of points occurring on a straight line or analogously as events occurring through time (Fig. 1, and see Table 1 for glossary of terms).

## Introducing the Poisson process

A commonly used point process is the Poisson process (*Daley & Vere-Jones, 2003*). It has a single parameter, the arrival rate ($\lambda$) for a given unit of time (*e.g.*, 10 arrivals/hour). By using counts of behaviour (from a sampling period) as a representative measure of that behaviour, we are making the assumption that there is an underlying rate, which we try to capture during the observation. As we want to understand the factors affecting the rate at which bees enter a nest, it is this parameter, $\lambda$, that we are broadly interested in estimating (note that we never observe this rate directly, it is an underlying, latent property of this process).

The simplest Poisson process is the homogeneous Poisson process, which assumes that the arrival rate ($\lambda$) is constant through time. Given the simplicity of this process, we believe that it is a highly suitable starting point or null model with which to describe behavioural counts. It also has several useful properties, which we can compare with real data to assess the suitability of this model (explored in *Does a Poisson process provide a good description of behaviour?*). First, it assumes that the probability of arrival is constant over time, which results in the interval lengths within an observation being exponentially distributed. Second, a Poisson process results in a predictable amount of variation in the number of visits observed (per observation) across multiple observations; these counts (for each observation) follow a Poisson distribution, with mean (and variance) equal to the length of the observation period ($t$) multiplied by the arrival rate, or $\lambda t$. We will describe this mean as the *expected* number of arrivals for an observation (c.f. *de Villemereuil et al., 2016*). In other words, for every observation there is an underlying arrival rate leading to an expected number of arrivals, but we observe this with a certain degree of (quantifiable) error caused by the stochastic nature of the process (Fig. 2A). For example, if $\lambda$ is 10 arrivals/hour, then the expected number of arrivals in one hour would be 10, two hours 20 etc, but we would observe considerable variation around these expectations. Formulaically we describe this as $y \sim Poisson(\lambda t)$, where y is the observed counts. We will refer to this error as *stochastic* error. In the statistical literature, this would be referred to as aleatory uncertainty (*Kiureghian & Ditlevsen, 2009*). This error is analogous to measurement error, with similar consequences (outlined below).

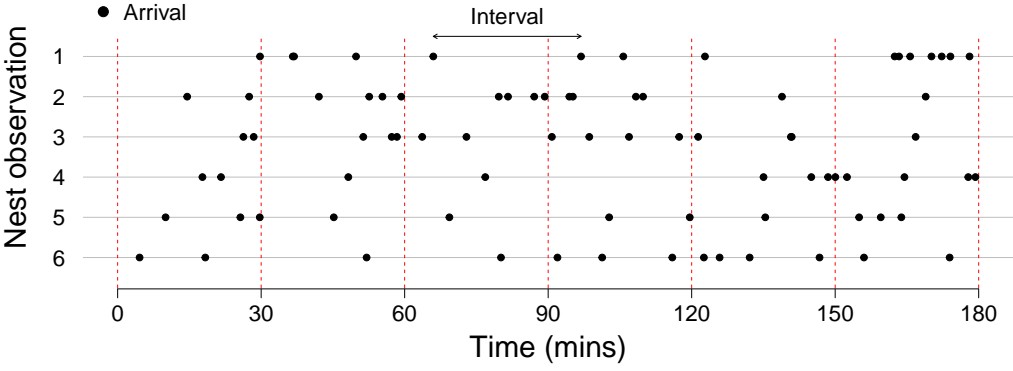

**Figure 1** **Behaviour can be described as points (or events) occurring on a straight line (through time), in other words as a point process.** Six nest observations are shown (grey lines), that were all simulated using a Poisson process with the *same* arrival rate (4.5 arrivals/hour) to demonstrate the variation that may arise through observations of different nests with the same arrival rate. The red dotted lines demonstrate the effect of shortening observation periods on the variation between observations.

**Table 1** **Glossary of terms used in the manuscript.**

| | |
|---|---|
| **Point/event** | Occurrence of a behaviour |
| **Rate** | Number of events per unit time |
| **Observation** | Observation of the unit of interest (*e.g.*, an individual or a nest) for a defined period of time |
| **Observation period** ($t$) | The length of the observation (*e.g.*, one hour) |
| **Interval** | The interval between two events (*e.g.*, arrivals at a nest) |
| **Interval length** | The length of time between two events |
| **Point process** | Statistical description of events occurring through time |
| **Poisson process** | Simple point process with a single parameter, the rate ($\lambda$). |
| $\lambda$ | 'True' rate, an underlying/latent, unmeasureable variable. Equal to 1/expected interval length. When we use the number of arrivals in an observation period or the mean interval length, we are implicitly estimating this quantity. |
| **Expected number of events** | $\lambda t$ *i.e.*, the number of events we would *expect* to see in a given time period and a given occurrence rate |
| **Observed number of events** ($y$) | Number of events actually *observed* in an observation period |
| **Stochastic error** | Error induced in our estimate of rate through sampling design |
| **Refractory period** | A period in which a behaviour is unlikely to reoccur |

Now imagine that we want to compare different bee nests (for instance we may be interested in the differences in arrival rate due to factors such as the nest's size, distance to food *etc.*). If all nests had the same arrival rate, then by watching them for the same observation period, the observed number of arrivals across these observations would be Poisson distributed (Fig. 2A). In other words, we would still observe variation in the number of arrivals between observations of different nests, even if there is no variation in
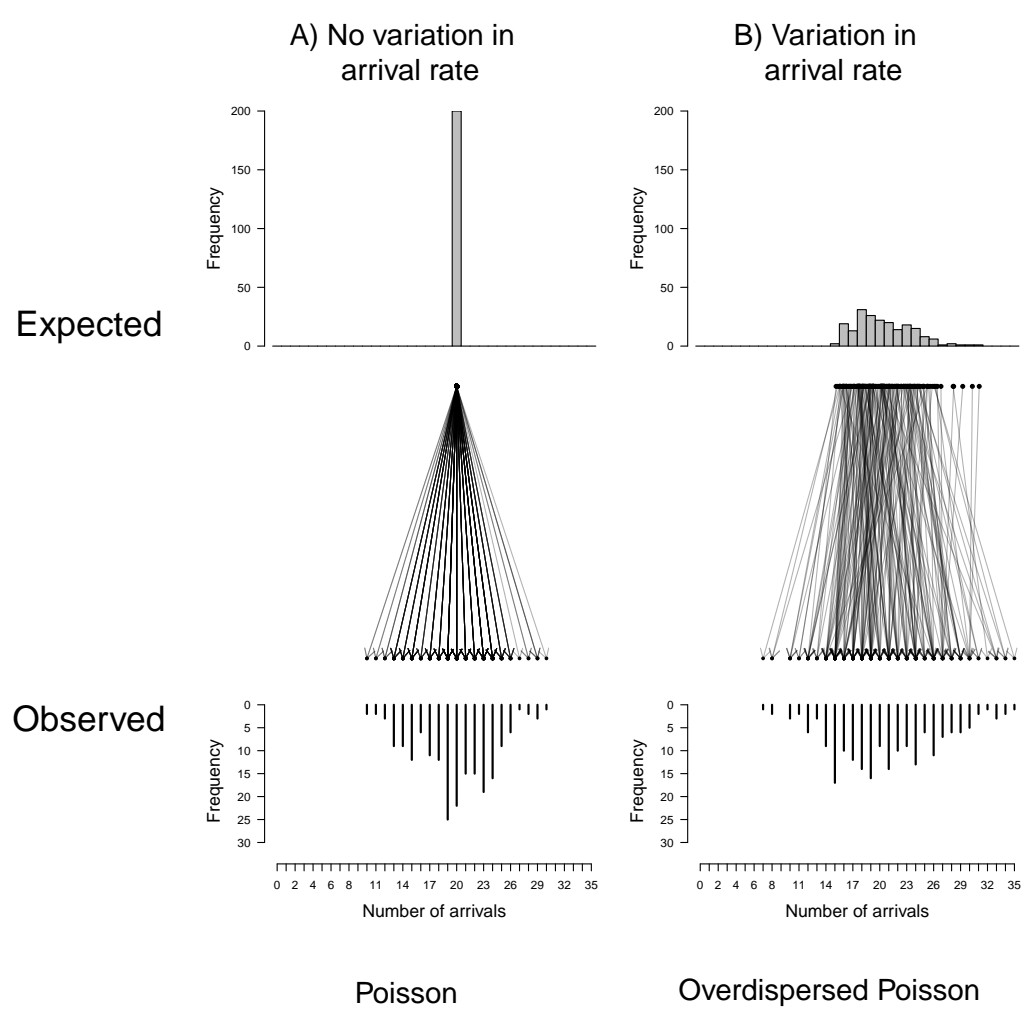

**Figure 2  Visualisation of Poisson distributed stochastic error.** (A) If all observations are the same length and have the same expected rate (*e.g.*, $t = 2$, $\lambda = 10$ and $\sigma^2_{\lambda t} = 0$, so $\lambda t = 20$), the number of visits across all observations would be Poisson distributed ($\sigma^2_{stoc} = 20$). (B) When there is variation in the expected rate (for example, due to consistent differences between individuals; $\sigma^2_{\lambda t} > 0$), every different rate is observed with stochastic error, leading to an over-dispersed Poisson distribution on the observed scale.

their arrival rates (Fig. 2A). If the arrival rates differ between nests (*i.e.,* variation in the expected number of arrivals, due to factors such as nest size *etc.*) the combined observations would be over-dispersed (*i.e.,* more variation present than explained simply by the Poisson distribution; Fig. 2B).

## Diminishing stochastic error

As mentioned above, it seems intuitive that longer observations result in lower error (*Lendvai et al., 2015*; *Morvai et al., 2016*). As both the mean number of observed arrivals across all observations increases (for example through longer observations) and with greater variation in arrival rates, the amount of stochastic error, relative to total observed variance, diminishes. Let's look more carefully at why this is the case.
First, following the law of total variance, the total observed variance in the number of arrivals across observations ($\sigma_y^2$) is equal to the expectation of the stochastic variance ($\sigma_{stoc}^2$) plus the variance in expected number of arrivals ($\sigma_{\lambda t}^2$)

$$\sigma_y^2 = \sigma_{stoc}^2 + \sigma_{\lambda t}^2. \tag{1}$$

Second, the stochastic variance and variance in the expected number of arrivals do not scale in the same way. This is most easily demonstrated by thinking of variability in terms of the coefficient of variation, a dimensionless measure of variability ($CV_x = \frac{\sigma_x}{\bar{x}}$, where $\bar{x}$ and $\sigma_x$ are the mean and standard deviation, respectively, of $x$). The variation in expected number of arrivals, when measured as CV, remains constant as the mean number of arrivals increases. This is because the standard deviation scales directly with the mean (*e.g.*, $\sigma_{2x} = 2\sigma_x$). So, if the observation period was twice as long, all the expected number of visits across observations would double, as would their SD, whilst the CV remains the same (Fig. 3A; $\frac{2\sigma_x}{2\bar{x}} = \frac{\sigma_x}{\bar{x}}$)). Put as an equation, if the observation period ($t$) is constant across observations,

$$CV_{exp} = \frac{\sigma_{\lambda t}}{\bar{\lambda} t} = \frac{t\sigma_\lambda}{t\bar{\lambda}} = \frac{\sigma_\lambda}{\bar{\lambda}} \tag{2}$$

and so changing the observation period does not change the CV in the expected number of visits. In contrast, due to the nature of the Poisson distribution (*i.e.*, the mean equals the variance), as the mean (number of arrivals) increases, the Poisson distributed stochastic error becomes relatively less variable (*i.e.*, CV decreases; $CV_{stoc} = \frac{\sqrt{\bar{\lambda} t}}{\bar{\lambda} t}$; Fig. 3A). In other words, the stochastic error diminishes relative to the variation in expected number of arrivals as the sampled number of arrivals becomes larger.

Given that $\sigma_{stoc}^2 = \bar{\lambda} t$ and that $\sigma_{\lambda t}^2 = (\bar{\lambda} t CV_{exp})^2$ we can rewrite Eq. (1) as:

$$\sigma_y^2 = \bar{\lambda} t + (\bar{\lambda} t CV_{exp})^2. \tag{3}$$

We can now see that the variation due to stochastic error is equal to the mean, whilst the variation in the expected number of visits across observations is a function of both the expected CV (a constant) and the square of the mean. Therefore, as the mean number of observed visits increases, the variance due to the expected number of arrivals increases exponentially compared with stochastic error. This results in a rapid decrease in the observed CV and an increase in the proportion of the observed variation due to variation in the expected number of arrivals (Figs. 3B, 3C). In other words, the amount of times a behaviour occurs in an observation period directly affects the confidence we can have in quantifying the rate at which it occurs. Equally, the greater the amount of underlying variation in arrival rates (*i.e.*, expected CV; shown by different lines in Figs. 3B, 3C), the lower the impact the stochastic error has and the more the observed variation reflects this expected variation in arrival rate (Fig. 3C).

This stochastic error can be seen as analogous to measurement error; if we take a variable measured with a large amount of error, averaging over an increasing number of measurements would give a better estimate. Similarly, the more events we observe, the more the stochastic error is averaged over, and the greater precision we have in our estimate of λ.
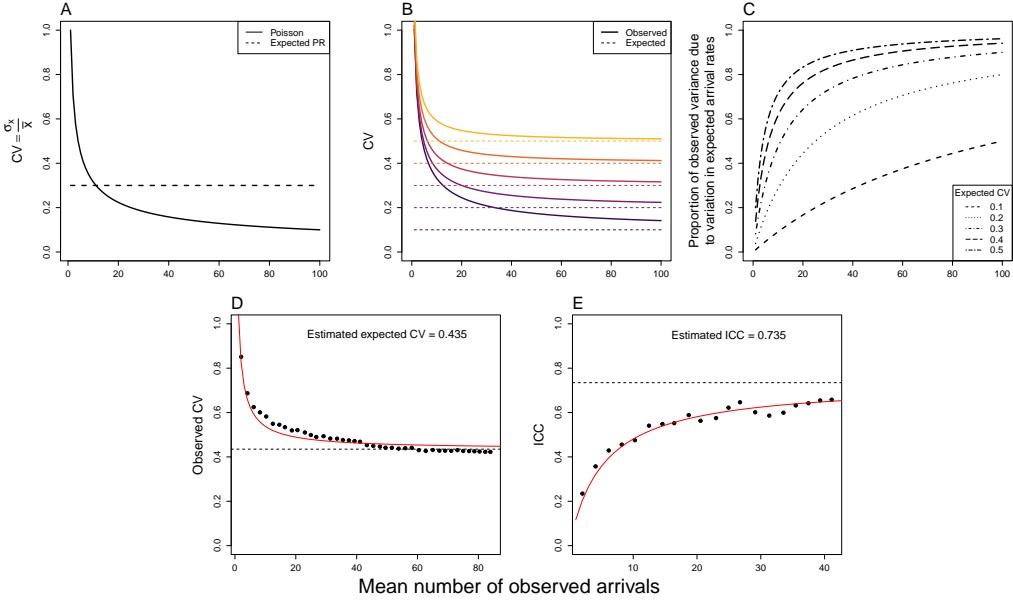

**Figure 3** **In data with Poisson distributed stochastic error, we expect to see certain patterns.** Because the coefficient of variation (CV) of Poisson distributed stochastic error (solid line) and expected arrival rate (dotted line), change differently as the mean number of observed arrivals increases (A), we see that the observed CV (solid line) decreases as the mean number of observed arrivals increases, according to the variation in expected rates (B). In (D) we can see this pattern in real provisioning data. As a result, the proportion of total observed variation due to 'biological' variation in expected rates, should increase with the mean number of observed arrivals (C). This proportion can be interpreted as either the maximum repeatability or $R^2$ when behavioural counts are analysed as a response variable or the maximum amount of variation these counts can explain in another variable, when Poisson distributed stochastic error is not accounted for. In E), we see this pattern arising in real provisioning data; repeatability (ICC) increases with increasing observation period. (D and E) use data presented in *Lendvai et al. (2015)*; the red line shows the predictions from a non-linear model estimating CV and ICC, respectively, assuming an underlying Poisson process, and dotted line the estimated CV and ICC from these models (see Supplementary Material S2).

## Dynamic rates

A homogeneous Poisson process assumes that the rate at which a behaviour occurs is the same throughout the observation. It is, however, possible that individuals adjust their behaviour at a very fine scale (*e.g. Johnstone et al., 2014*; *Schlicht et al., 2016*), meaning that the expected rate changes during the observation. A changing arrival rate does not, however, violate the assumptions of a general Poisson process. Indeed, there exist models that allow the rate to be dynamic (*e.g.*, inhomogenous Poisson processes *Heuer, Mueller & Rubner, 2010*). For example, in a Cox point processes, a generalisation of the Poisson process, the arrival rate is also assumed to be stochastic, and modelled as a latent variable (*Blackwell et al., 2016*; *Spence et al., 2021*; see also *Johnstone et al., 2014* for a related example in behavioural ecology). When not modelled, such dynamically changing behaviour will add further error to the estimation of the overall rate at which a behaviour occurs. Consequently, if the behaviour of interest is extremely dynamic, we would not expect to find anything we measure on a broad scale to correlate with it. However, many behaviours have been

found to consistently differ between individuals, as shown in the recent flurry of studies on animal personality (*Bell, Hankison & Laskowski, 2009*; *Beekman & Jordan, 2017*), and are frequently found to vary among different environments. Such findings indicate that there is a consistent rate that can be measured over these time periods in many behaviours.

## Refractory periods and the non-independence of visits

A homogeneous Poisson process also assumes that, at any time point, the time to next arrival is independent of when the previous arrival occurred (*i.e.,* the likelihood of an event occurring is constant over time). This is known as the Markov property, and results in an exponential distribution of interval lengths (with a modal interval length of 0). Up until now we have been considering arrivals of bees at a nest, which are likely to be largely independent of each other. When considering the behaviour of a single individual, however, this may seem unrealistic. It is important to note that when considering this assumption of independence, it does not matter if we are watching an individual, a pair, a group or a set of unique individuals. The assumption of independence relates simply to whether the timing of one event affects the occurrence of the next event, and not to the individual that does the event (although the chance that the assumption of independence is violated may increase with fewer individuals).

Let us consider then, the arrival rate to the nest of an individual bee (rather than the overall arrival rate at the nest). As bees have to find food and return to the nest, the probability of this individual bee arriving at the nest is likely to be lower just after its last arrival occurred, meaning the probability of the bee arriving changes over time. This is known as a refractory period (*i.e.,* a period in which a behaviour is unlikely to reoccur). A refractory period can be described in a point process with an additional parameter. As stated above, a Poisson process assumes that the interval lengths follow an exponential distribution, with a modal interval length of 0. The exponential distribution is a special case of the gamma distribution, in which one of its two parameters, $\alpha$, is fixed to 1 (no refractory period). When $\alpha > 1$, we have a point process with a refractory period. $\alpha$ describes the refractory period in terms of the expected interval length. The refractory period itself can be more intuitively thought of as the mode of the gamma distribution (see Supplementary Material S1).

As the interval lengths no longer follow an exponential distribution when there is a refractory period, the number of arrivals in a given period of time would no longer follow a Poisson distribution, violating the assumptions of the Poisson process. A refractory period reduces the amount of stochastic error for a particular mean (*i.e.,* underdispersion with respect to a Poisson distribution) in a predictable way, and so can be modelled using the additional $\alpha$ parameter to describe the relative extent of the refractory period (see Supplementary Material S1). Although a refractory period results in less stochastic error for given mean number of observations, it is important to note that the stochastic error will always be proportional to this mean, and so the relative amount of stochastic error will still diminish as more events are observed, or with more biological variation. In the context of the equations above, with a refractory period $\sigma^2_{stoc}$ becomes $\bar{\lambda}t/\alpha$ rather than $\bar{\lambda}t$ ($1/\alpha$ is commonly refered to as $\phi$ or the dispersion parameter in some analytical models).

Furthermore, small refractory periods do not lead to substantial deviations away from a Poisson process.

In behavioural data, there are several instances where such refractory periods may exist, for example, if an individual has to do something before the behaviour reoccurs. As in the example above, this might be seen in arrival or feeding rates, especially if foraging sites are located far away from the nests. Sexual behaviours are also likely to show such refractory periods; this is well studied in rats and humans for example (*Levin, 2009*), and it is likely that there is a minimum interval length between copulations in many other species. Quantification of refractory periods is therefore important, but to our knowledge has not yet been systematically investigated for any behavioural trait measured this way (at least in the context of behavioural ecology). The little quantitative information about the presence, and length, of refractory periods makes it very difficult to judge their impact, at least currently.

### Embracing a null model

*… there is no need to ask the question "Is the model true". If "truth" is to be the "whole truth" the answer must be "No". The only question of interest is "Is the model illuminating and useful?"* George Box, 1979

Here we argue that a Poisson process is a good *null* model to describe the stochastic nature of behaviours sampled in specific periods of time (a type of point process) for the following three reasons. First, it is simple and tractable, and it allows us to account for how stochastic error predictably changes with sampling effort. We acknowledge that it may not be the perfect model for such behaviours in all circumstances (as discussed above). However, this is not the purpose of a null model. Currently, we have no clear null model, as generally we simply ignore the presence of this stochastic error (see literature survey below), which is induced by the processes underlying behavioural count data. As we will discuss below, ignoring this stochastic error (the presence of which we believe to be undeniable, the form/extent of which can be the subject of debate) leads to systematic error in the analysis of behavioural counts. Second, deviation from this model gives us valuable information. We believe that assuming (and more importantly understanding) such a null model gives us insight into the processes underlying the data, whether deviations from this model occur, and if so what they may represent. Many fields have embraced null models (for example, the ideal-free distribution in behavioural ecology or the Hardy-Weinberg Equilibrium in population genetics), and it is standard practice to quantify deviation from these models. Finally, such a null model would force us to confront what assumptions we are making when analysing our behavioural count data.

## PROBLEMS WITH IGNORING STOCHASTIC ERROR

Generally, stochastic error induces the same analytical problems as measurement error. Although these have been covered elsewhere (*e.g.*, *Freckleton, 2011*; *Garamszegi, 2016*; *Ponzi et al., 2018*; *Dingemanse, Araya-Ajoy & Westneat, 2021*), they have not been discussed in the context of behavioural count data, and so we briefly outline the problems here for completeness.

## Analysing variation in behavioural counts

When stochastic error is not explicitly modelled, it is included in the residual, unexplained variation (Fig. 4). This imposes an upper limit to the variance in, for example, arrival rate that can be explained (Fig. 3C), because the stochastic error will always remain unexplained, unless explicitly accounted for (Fig. 4; *Nakagawa & Schielzeth, 2013*; *Nakagawa, Johnson & Schielzeth, 2017*). To demonstrate this, let us return to our population of bee nests where the mean (±SD) arrival rate across nests is $10 \pm 3$ arrivals/hour (*i.e.,* an expected CV of 0.3), and 50% of this expected variation in arrival rate is due to consistent differences between nests (*i.e.,* repeatability (ICC) on the expected scale $= 0.5$). Assuming that arrival rates in this population are well described by a Poisson process, a study that observed nests for 2 h (Study A; Fig. 4), would observe an average of $20 \pm 7.5$ arrivals per observation and estimate a repeatability of 0.32, just over half of the actual repeatability. Not accounting for this stochastic error leads, therefore, to a general underestimation of underlying effect sizes.

Different studies will also vary in the mean number of observed arrivals and/or the underlying variation in expected arrival rates, through having different observation periods, or simply because of intrinsic differences among populations. As the proportion of total variance due to stochastic error is dependent on both of these factors (Fig. 3C), metrics that relies on the estimation of total variance (*e.g.,* standardised effect sizes, ICC and $R^2$) are not comparable between studies, when not accounting for this changing stochastic error. Imagine that two more studies (Studies B and C) are performed in the population described above, but with shorter observation periods (60 and 30 mins, respectively), averaging 10 and 5 arrivals per observation. As a different amount of stochastic error was observed in both cases, the resulting repeatabilities would be much lower still, 0.24 and 0.16 respectively (Fig. 4). Effect sizes, therefore, may differ between studies due to both the intrinsic characteristics of the population and the sampling effort. Note that these calculations assume an underlying Poisson process. If, for example, the refractory period was to differ between different studies then, without accounting the differing form of stochastic error, the results would also not be comparable.

## The low predictive power of behaviour

Behavioural count data typically correlates poorly with other variables. However, because of the potentially large proportion of observed variation that is due to stochastic error, the observed number of events is constrained in how much variation in another trait it can explain (Figs. 3C, 4). For example, arrival rate estimated from a short observation period will correlate poorly with the underlying arrival rate, due to this stochastic error (*Lendvai et al., 2015*; *Morvai et al., 2016*). Consequently, this measure may explain little variation in another variable—even if it actually has had a strong effect. Moreover, stochastic error in one predictor variable can have a large effect on the parameter estimates of other covariates in the model, as the covariance between different parameters is not properly estimated, creating potentially spurious relationships between predictor variables and the response variable (*Freckleton, 2011*). Note that these effects are a general consequence of any kind

**Example Population**
Arrival Rate = 10 visits/hour, Expected CV = 0.3, Repeatability (ICC) = 0.5

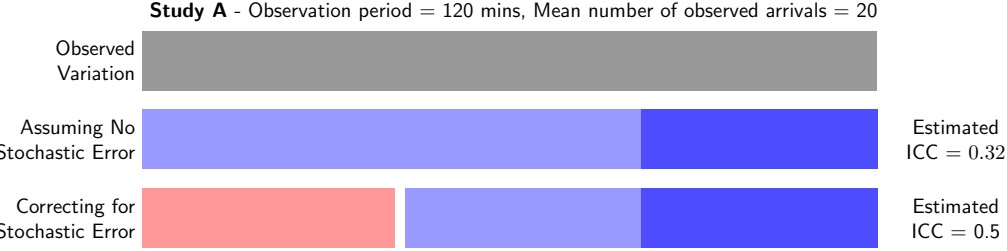

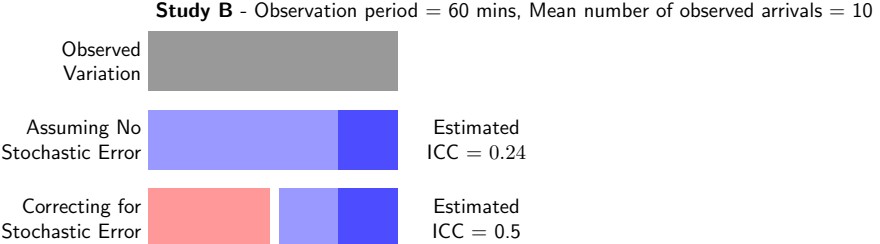

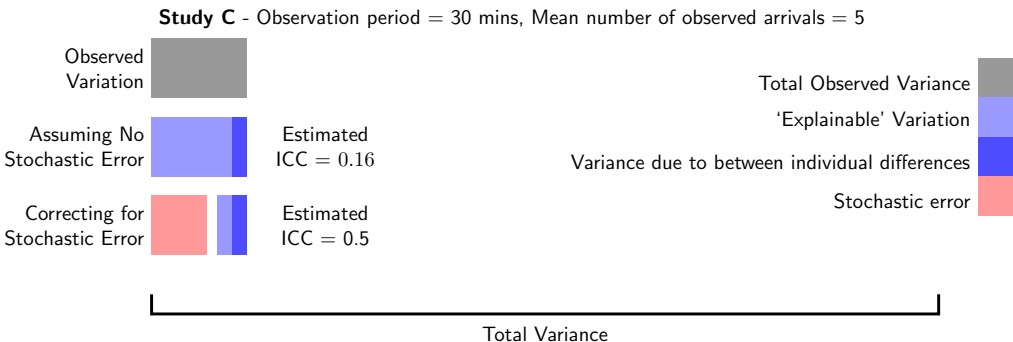

Total Variance

**Figure 4** **Effect of not accounting for Poisson distributed stochastic error in analyses of behavioural count data.** Three studies of different observation periods in the same population will estimate different effect sizes, when not accounting correctly for the presence of stochastic error.

of measurement error in predictor variables, but will be particularly pronounced with the level of error seen in count data.

## DOES A POISSON PROCESS PROVIDE A GOOD DESCRIPTION OF BEHAVIOUR?

Up until now we have been focusing on a general example whilst arguing that a Poisson process represents a suitable descriptive model for behavioural count data. Now we will take

a frequently used behavioural count—provisioning rate—to demonstrate the utility of this model and highlight the extent of the problems caused by not taking such stochastic error into account. Parental provisioning rate (measured as the number of feeding visits within a certain unit of time) is often used as a quantitative assessment of parental investment in birds and analyses of provisioning rate have contributed a considerable amount to our understanding of parental care (*e.g. Harrison et al., 2009*). As a Poisson process makes certain assumptions, we can compare the patterns we see in observed data with those we expect from a Poisson process, to assess how good a model this is for describing our data. There are three important patterns which can emerge.

First, a Poisson process assumes that visits to the nest are independent from each other, in others words there is no refractory period. Note that the assumption of independence is not violated by our watching the same individual, rather referring to the probability of a visit occurring depending on when the last visit occurred. Although this is perhaps the most obvious way in which provisioning rate could violate the assumptions of a Poisson process, at the moment, there is little evidence that substantial refractory periods exist for provisioning rate. For example, distributions of inter-visit interval lengths that appear close to exponential (as expected from a Poisson process) have been observed in several species (Great tit (*Parus major*)—*Johnstone et al., 2014*; Acorn Woodpecker (*Melanerpes formicivorus*)—Figure 3 in *Koenig & Walters (2016)*; Red Winged Blackbird (*Agelaius phoeniceus*)—Figure 2 in *Westneat, Schofield & Wright, 2013*; Pied Flycatcher (*Ficedula hypoleuca*)—Figure S1 in *Westneat et al., 2017*; Chestnut-crowned Babbler (*Pomatostomus ruficeps*)—*Savage et al., 2017*; House Sparrow (*Passer domesticus*)—Figure S1 in *Ihle et al., 2019*). Furthermore, many studies analyse per nest visit rates (*i.e.,* with two or more parents/carers), in which case refractory periods are likely to be extremely low.

Second, from a Poisson process we would expect Poisson distributed error. With additional factors influencing provisioning rate (due to individual, brood or environmental characteristics), we would see additional variation (*i.e.,* overdispersion with respect to a Poisson distribution; Fig. 2) and so the variance in the number of observed visits between observations would be greater than the mean. Observing overdispersion does not necessarily mean that the stochastic error is Poisson distributed; we could still observe more variation than expected in a Poisson distribution, if for example the stochastic error was lower (due to a refractory period) and biological variation greater. The variance being more than or equal to the mean is rather a minimum requirement for Poisson distributed error to exist. Consistent underdispersion with respect to a Poisson distribution across population level estimates of variation, on the other hand, would indicate that the error was not Poisson distributed, violating the assumptions of our null model. As shown below (see Literature Survey), overdispersion is consistently found across studies and species in provisioning rate, which is consistent with our model of a Poisson process and additional between-observation variation in expected provisioning rates.

Finally, we would expect to see a dramatic decrease in the relative variability of provisioning data with an increase in the mean number of observed visits (Fig. 3A). Using recently published data (*Lendvai et al., 2015*), we can see this systematic decrease in CV with increasing observation time (Fig. 3D; see Supplementary Material S2 for further

details), in accordance with theoretical predictions. Concurrently we would also expect that metrics such as repeatability would increase with the mean number of observed visits (Fig. 3C). Again, we can show this effect empirically using the data of *Lendvai et al. (2015)*. By calculating the repeatability of provisioning rate using the same overall total time period, but split into differently sized observations periods (and not correcting for this stochastic error), we indeed find that repeatability dramatically increases with observation period (*i.e.,* with mean number of observed visits; Fig. 3E, Supplementary Material S2), in line with expectations from a Poisson process.

Together this evidence demonstrates that provisioning data has a high level of stochastic error, the magnitude of which is in line with that predicted by a Poisson process. Therefore, a Poisson process seems a highly suitable model for provisioning rate. We should stress, however, that the validity of these assumptions should be assessed on a behaviour and study specific basis.

## ASSESSING THE SIZE OF THE PROBLEM—LITERATURE SURVEY

Recent work has suggested that only a small amount of variation in provisioning rate is generally explained by individual, brood or environmental characteristics (*Williams, 2012*; *Williams & Fowler, 2015*), a trend that is commonly found across behavioural traits (*e.g.*, low repeatabilities; *Bell, Hankison & Laskowski, 2009*; *Wolak, Fairbairn & Paulsen, 2012*). It has also been suggested that provisioning rate often has little or no detectable effects on offspring phenotype (*e.g.*, fledgling size, survival etc; *Schwagmeyer & Mock, 2008*; *Williams, 2012*; *Williams & Fowler, 2015*), bringing into question its utility as an indicator of parental investment. However, such conclusions may arise from failing to account for the presence of stochastic error. How much of a problem these issues present depends largely on both the sampling effort employed in such studies, how variable this effort is among studies, and whether or not the presence of stochastic error is accounted for. In order to ascertain the breadth of the problems outlined above, we conducted a survey of papers analysing provisioning rate (measured as the number of visits), published in 2015/16. We did not intend the search to be exhaustive, rather to generate a representative set of recent papers on provisioning rate.

### Survey methods

On 13/12/2016 JLP searched Web of Science using the search term:

```
(TS=("visit rate" OR "number of visit*" OR "nest visit*" OR "provisioning"
OR "feeding rate" OR "parental care" OR "number of feed*") AND TS=(chick*
OR nest* OR fledg* OR offspring) AND TS=(*bird* OR passerine* OR avian
OR chick*) NOT TS=(veterinary OR chicken* OR hen* OR broiler* OR poult*
OR layer* OR "Japanese quail*" OR turkey* OR chickpea* OR pollin*)) AND
PY=(2015 OR 2016) AND LANGUAGE: (English) AND DOCUMENT TYPES: (Article)
```

returning 289 papers. JLP and MI screened the abstracts to exclude any papers that did not relate to provisioning (or show primary data *e.g.*, reviews). JLP and NK read the remaining 143 papers, looking specifically for studies that measured provisioning as number of visits, as opposed to inter-visit intervals or quantity of food. Although we did

not target these studies, we included any studies that measured incubation feeding by males. This process returned 81 studies. At a reviewers request, JLP repeated this process for papers published in 2022, searching Web of Science on 05/12/2022, returning 147 papers, which reduced to 46 after abstracts screening, 29 of which analysed provisioning data.

From the 81 papers from 2015/16, JLP and NK extracted the study species, the method of data collection (direct, video, RFID *etc.*) and length of observation period (hours). We also extracted summary data for each analysis that was conducted using provisioning rate, totalling 427 analyses across all the studies. For each analysis, we recorded whether provisioning was used as a response or predictor variable. If analysed as a response, we recorded what error distribution was used *e.g.*, Gaussian (including linear regressions, t-tests, ANOVAs, correlations), Poisson, Negative Binomial, non-parametric etc. If not otherwise stated, we assumed Gaussian error distribution was used, because this is the default in most statistical software. We recorded whether the number of visits itself was analysed, or whether it was transformed into a rate (*e.g* visits per hour or visits per chick). The latter of these metrics (response/predictor, distribution, number/rate) were also extracted from the 29 papers from 2022 by JLP, giving 79 analyses.

For each 2015/16 analysis, we then extracted, where possible, mean and SD of provisioning data used in that analysis (note some analyses used the same data, so we recorded how many times that data set was used and in what ways). If SE and sample size (N) were presented SD was calculated as $SD = SE\sqrt{N}$. If range and/or inter-quartile range were presented then a SD was derived following the formulas presented in *Wan et al. (2014)*. If not presented in the text, mean and SD or SE were extracted from figures using the *digitize* and *metaDigitise* packages in R (*Poisot, 2011*; *Pick, Nakagawa & Noble, 2019*), or from raw data if available. If parameter estimates from models were presented, the intercept was taken as a measure of the mean provisioning rate (and back-transformed if necessary), as long as any covariates were centered, but the associated SE was not used. We aimed to collect means and SDs that were representative of the data that was analysed. This meant that when means and SDs from different groups included in the same analyses were presented(for example, means presented per sex, but data analysed across both sexes), we pooled these $M$ estimates (from samples $x_1$ through $x_M$) according to the formulas:

$$\bar{x} = \frac{\sum_i N_{x_i} \bar{x}_i}{\sum_i N_{x_i}} \tag{4}$$

$$\sigma_x = \sqrt{\frac{1}{\sum_i N_{x_i} - 1}\left(\sum_i \left[(N_{x_i}-1)\sigma_{x_i}^2 + N_{x_i}\bar{x}_i^2\right] - \left[\sum_i N_{x_i}\right]\bar{x}^2\right)}. \tag{5}$$

Where the mean (±SD) provisioning was presented as a rate (*i.e.*, had been transformed from the scale on which it was collected), we back-transformed these to their original scale, *e.g.*, if the observation period was 4 h, but provisioning was presented as visits/hour, we multiplied both the mean and SD by 4. For estimates where the

observation period varied, we corrected them relative to the mean observation period (or mode if this was presented). Similarly, when rates were presented as per chick, and the mean number of chicks was presented, then we back calculated the mean, but not SD. For estimates for which we had SD, and were not presented as per chick, we calculated the expected CV as

$$CV_{exp} = \frac{\sqrt{\sigma_x^2 - \bar{x}}}{\bar{x}}. \tag{6}$$

For several estimates, especially when the mean is low the expected CV cannot be calculated, and the mean is larger than the variance (see Supplementary Material S5). To these estimates we gave a value of 0. We then calculated the proportion of observed variation due variation in expected provisioning rate as

$$\frac{\bar{x}CV_{exp}^2}{1 + \bar{x}CV_{exp}^2}. \tag{7}$$

The papers included in the study, and the data extracted from them, are presented in Table S1.

## Survey results

Our survey of papers published in 2015/16 found 81 studies of 64 species that fitted our criteria, containing 427 analyses of provisioning rate (343 as a response and 84 as a predictor) and our 2022 survey found 29 studies containing 79 analyses of provisioning rate (50 as a response and 29 as a predictor). For 350 of the 2015/16 analyses we could extract the mean number of observed visits for the data used in the analysis, which was typically low (median = 8.41, Fig. 5). For 301 analyses we were also able to extract a standard deviation of provisioning rate, and so calculate expected CV. For 34 of these analyses, it was not possible to calculate expected CV, and so it was assumed to be 0 (Supplementary Material S5). Expected CV ranged from 0 to 1.234 (median =0.449).

Using the expected CV and mean number of observed visits for each sample, we were able to calculate the proportion of observed variation in provisioning rate that is due to expected, biological, variation (see Supplementary Material S3). Across these estimates, the median proportion was 0.627 (Fig. 5). This represents the maximum effect size (*e.g.*, $R^2$ or ICC) that can be estimated from this data, if sampling error is not accounted for (*i.e.,* if the true ICC was 1 the estimated ICC would be 0.627, and if the true ICC were 0.5, then the estimated ICC would be 0.3135 etc.). Of the 427 analyses, only 7% and 22% (in 2015/16 and 2022, respectively) modelled the stochastic error by assuming a Poisson error distribution when provisioning rate was analysed as a response variable (see below, note no other corrections were used), whilst no study in either survey accounted for stochastic error when modelling provisioning rate as a predictor variable. Effect sizes from these studies are therefore consistently, and often substantially, underestimated. Repeatabilities, for example, would be underestimated on average by 37%, as would the effect of provisioning rate on other variables such as chick mass or survival. Moreover, because of the large variation in this proportion among studies (range: 0–0.982), interpretation of, and comparison amongst, effect size estimates from these studies is not meaningful, as any

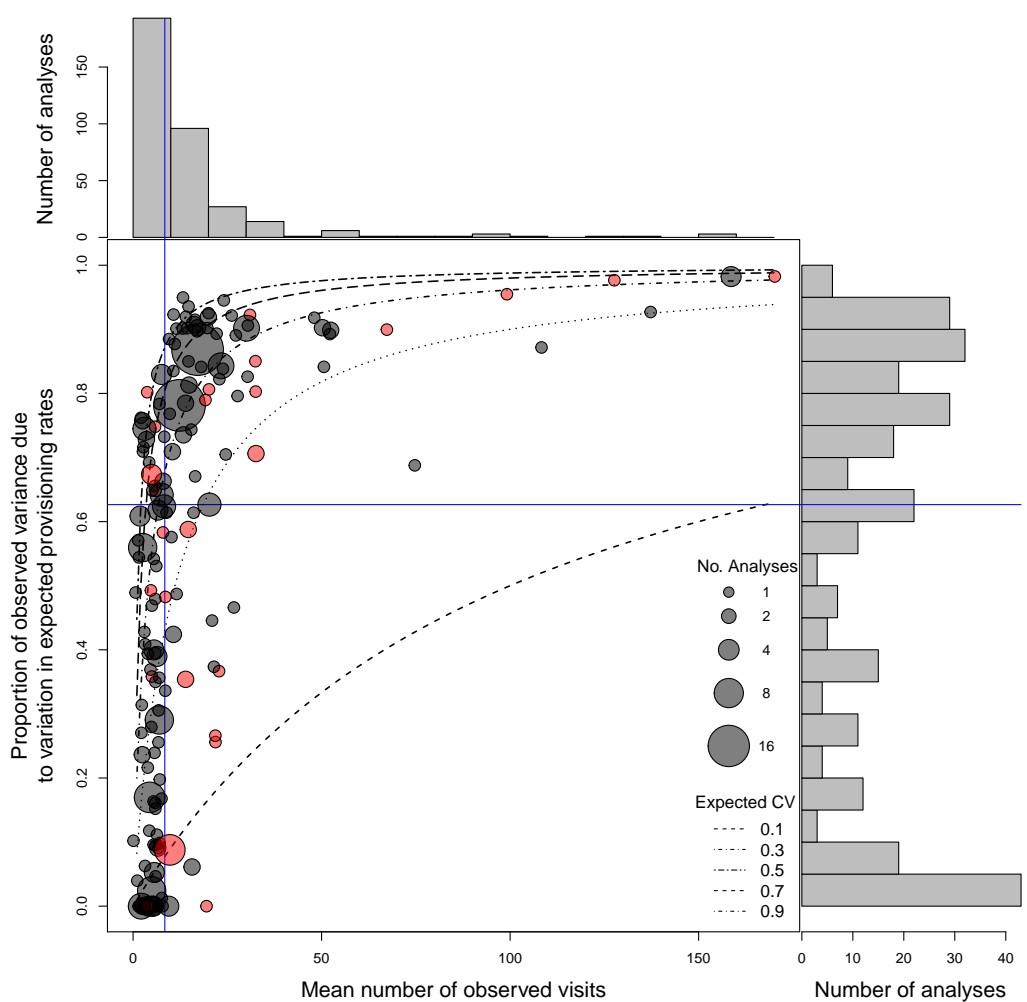

**Figure 5** **Distributions of the mean number of observed visits and proportion of observed variation due to expected variation in provisioning rates, from provisioning data used in analyses presented in papers published in 2015/16, and the relationship between them.** Points show individual datasets, the size of the points indicates the number of models that were run using this dataset. Grey points indicate data from direct observation or video recordings, and red from automated data collection. Blue lines show median values. Estimates for which the mean was greater than the variance, the proportion of observed variation is displayed as 0.

differences may simply be due to methodology, rather than to biological differences. More generally, no study on the repeatability of provisioning behaviour has accounted for (*i.e.,* removed) this sampling error (see *Khwaja et al., 2017* for summary of studies), suggesting that these estimates are underestimations, and that the comparison among these studies may be problematic. Note that we have assumed in these calculations that this stochastic error is Poisson-distributed. As discussed above, this may not be the case. However, this survey still demonstrates how wide ranging the current underestimation of effect size is. Furthermore, as refractory periods may vary between studies, not accounting for the stochastic error makes comparison between studies extremely problematic.

## ANALYTICAL SOLUTIONS

### Directly modelling stochastic error

In the majority of analyses (80% and 63% in our surveys of 2015/16 and 2022, respectively), behavioural counts are analysed as a response variable. Stochastic error can be accounted for in such analyses by using a generalised linear mixed model(GLMM) framework, specifying a Poisson error distribution. Broadly GLMMs account for distribution specific variance (*i.e.,* the Poisson distributed stochastic error) seen on the observed level, and transform the data from the expected scale onto the 'latent' scale, using a 'link' function (in this case typically the log function). The data is normally distributed on the latent scale, fulfilling linear model assumptions (see *Bolker et al., 2009*; *Nakagawa & Schielzeth, 2010*; *de Villemereuil et al., 2016* for a helpful breakdown of these models). We present worked examples in the Supplementary Material S6.

By using GLMMs, the stochastic variation is specifically modelled, and ICC and $R^2$ estimates can be made with or without the Poisson distributed stochastic error (*i.e.,* by including it or not in the calculation of total variance; Fig. 4). Traditionally $R^2$ and ICC are calculated with stochastic error (*Nakagawa & Schielzeth, 2010*; *Nakagawa & Schielzeth, 2013*). However, the overriding utility of this has recently been discussed, and it has been suggested that under some circumstances it is more appropriate to measure such metrics on the expected scale (*i.e* without stochastic error; see *de Villemereuil et al., 2016*, for a more general discussion of this in the context of heritability, with reference to both Poisson and Binomial stochastic error). In the case of variables such as provisioning rate (that are sampled in a fixed period of time), we would argue that these metrics should be estimated without the inclusion of this stochastic error in the estimation of total variance (Fig. 4), as this is dependent on the sampling effort, and so the metrics are more biologically meaningful on the expected rather than the observed scale. Note that in some cases the actual behavioural count, rather than the rate, is more relevant to a researcher's question, for example correlating the total number of feeds within a time period with the mass change of chicks within that same time period. In this case, it would not be appropriate to account for the stochastic error (*i.e.,* by following the methods proposed in *Nakagawa & Schielzeth, 2010*; *Nakagawa & Schielzeth, 2013*, see also *Nakagawa, Johnson & Schielzeth, 2017*), as the number of feeds rather than the underlying rate would be the variable of interest.

Typically, provisioning rate has been analysed assuming a Gaussian error distribution (*i.e.,* in linear (mixed) models (L(M)Ms); 77% and 70% of analyses in our surveys of 2015/16 and 2022, respectively). The implicit assumption in these analyses is that all the variance can be explained *i.e.,* that there is no stochastic error that will always remain unexplained (*Nakagawa & Schielzeth, 2013*; *Nakagawa, Johnson & Schielzeth, 2017* Fig. 4). There has been much recent debate about the relative merits of using Gaussian or Poisson error distributions to model count data (*OHara & Kotze, 2010*; *Ives, 2015*; *Warton et al., 2016*; *Morrissey & Ruxton, 2020*). Here we emphasise that this stochastic error needs to be accounted for (which is possible using either method; see Supplementary Material S6). Our recommendation for GLMMs is therefore specifically based on the explicit estimation

of stochastic error in these models, leading to an output that is more intuitive and easier to deal with. Using LMMs to estimate effect sizes, and post hoc removing stochastic error, can also result in estimates that are not bounded by 0 and 1 (the limits of ICC and $R^2$). Gaussian distributions are also unbounded, suggesting that observations below 0 are possible. A common alternative is to assume a log-Gaussian distribution (*i.e.,* through log-transformation of counts). Although this is bounded above 0, 0 is not a value that can exist, whilst being a possible value for observed data (*OHara & Kotze, 2010*).

The resulting Poisson GLMMs are highly likely to be overdispersed, as it is unlikely that a given set of predictors will explain all underlying variation. To account for this (additive) overdispersion, models should be run as mixed models, including an observation level random effect (*Hinde, 1982*). The estimate of variance from this observation level random effect can be used as the estimate for overdispersion(non-distribution specific) variance, analogous to the residual variance of a linear model. Note that some software (*e.g.*, MCMCglmm; *Hadfield, 2010*) explicitly does this by default. It is also possible to model such overdispersion in other ways, *e.g.*, by assuming a negative binomial error distribution. This can be parameterised as a Poisson-gamma mixture distribution (rather than the Poisson-log normal mixture typically used in Poisson GLMMs).

It is worth noting that the parameter estimates themselves (*i.e.,* estimates of intercepts, slopes and variance components) should not change between the different models (log-normal and Poisson). These effect sizes only differ when standardised by total variance (which is how metrics are typically compared between studies). In studies of repeatability, we therefore also advocate reporting $CV_B$ - coefficient of between individual variation (*Holtmann, Lagisz & Nakagawa, 2017*; see also *Dochtermann & Royauté, 2019*). $CV_B$ reflects a mean-standardised measure of the amount of between individual variation. It is independent of the method of analysis and the degree of stochastic error and so can readily be compared between studies, regardless of sampling effort. It is analogous to $CV_A$ (the coefficient of additive genetic variation; *Houle, 1992*), which was proposed to address similar issues of comparing additive genetic variation between studies. Both $CV_A$ and $CV_B$ were first proposed in the context of Gaussian traits, and their derivation for other distributions is more complex. Recently, *de Villemereuil et al. (2016)* derived a formulation of $CV_A$ for non-Gaussian traits, which holds also for $CV_B$. From a Poisson GLMM, $CV_B$ is equal to the standard deviation of the between individual effects on the latent scale. A demonstration of the calculation of $CV_B$ is presented in Supplementary Material S6.

Our assumption here is that the stochastic error is Poisson distributed, which may not be the case if, for example, substantial refractory periods exist. The reduction in stochastic error due to such refractory periods is also predictable, and can be modelled with a Tweedie distribution (see Supplementary Material S1) or Generalised or Conway-Maxwell Poisson distributions (*Lynch, Thorson & Shelton, 2014*). Alternatively, the length of intervals between behaviours can be modelled, which we discuss further below. Note that, with or without a refractory period, current methods still act to systematically underestimate effect sizes, as they do not account for stochastic error. It should also be noted that when refractory periods are small, assuming Poisson distributed error results
in less bias than assuming no stochastic error (see Supplementary material S1). The choice that researchers make in how to calculate effect sizes (*i.e.,* whether to remove stochastic error, and if so the magnitude of that error) should be clearly defined in studies, allowing assessment and future recalculation of relevant effect sizes.

Figure 6 demonstrates the effect of using different methods in the estimation of $R^2$ on simulated data (Supplementary Material S5). In Fig. 6A, we can see that by not correcting for this stochastic error (using linear models; black dots), $R^2$ would increase as the mean number of observed visits increases and would be systematically underestimated, as is seen in real provisioning data (Fig. 3E). Accounting for this error using Poisson GLMMs, results in (predominantly) unbiased estimates of $R^2$ (except at low expected CV and mean number of observed visits; Fig. 6C). The precision of these models is also affected by both the expected CV and the mean number of observed visits, with precision increasing as they both increase (Fig. 6D).

## Behavioural counts as predictors—Measurement error models

Although stochastic error can easily be modelled when behavioural counts are the response variables, commonly used statistical software do not allow for the inclusion of error in predictor variables (linear models, for example, assume that there is no error in the predictors). Indeed, no analysis in our literature survey (out of 84 in 2015/16 and 29 in 2022) accounted for this stochastic error when provisioning rate was used as a predictor variable. In order to model this error, we can use a class of models, known as measurement error or 'error in variable' models, that allow error in the predictor variables to be specified. These models are, however, complex to implement at present, although can readily be created in software such as Stan (*Carpenter et al., 2017*). Measurement error models act very similarly to GLMMs, by creating a latent variable (*i.e.,* expected provisioning rate) which is then used as a predictor in the main model. Variation in observation time between different observations can therefore easily be accounted for, as can variables that may differ between observations. For example, if a researcher wanted to analyse the effect of provisioning rate on chick mass at fledging, but provisioning rate had been measured at different brood ages or in different environmental conditions at different nests, this variation could easily be accounted for. In Supplementary Material S6 we present a practical example of such models in Stan (see also *Freckleton, 2011*; *Garamszegi, 2016*; *Ponzi et al., 2018*; *Dingemanse, Araya-Ajoy & Westneat, 2021* for examples of the consequences of measurement error and how to deal with it in ecology and evolution).

In Fig. 6B we demonstrate the effect of not correcting for this stochastic error when using behavioural rate as a predictor variable (black dots); as the mean number of observed visits increases, the predictive power of the behaviour increases, but is systematically downwardly biased. Measurement error models (red dots) account well for this Poisson error, but as with GLMMs, their precision is low when the mean number of observed visits is low.

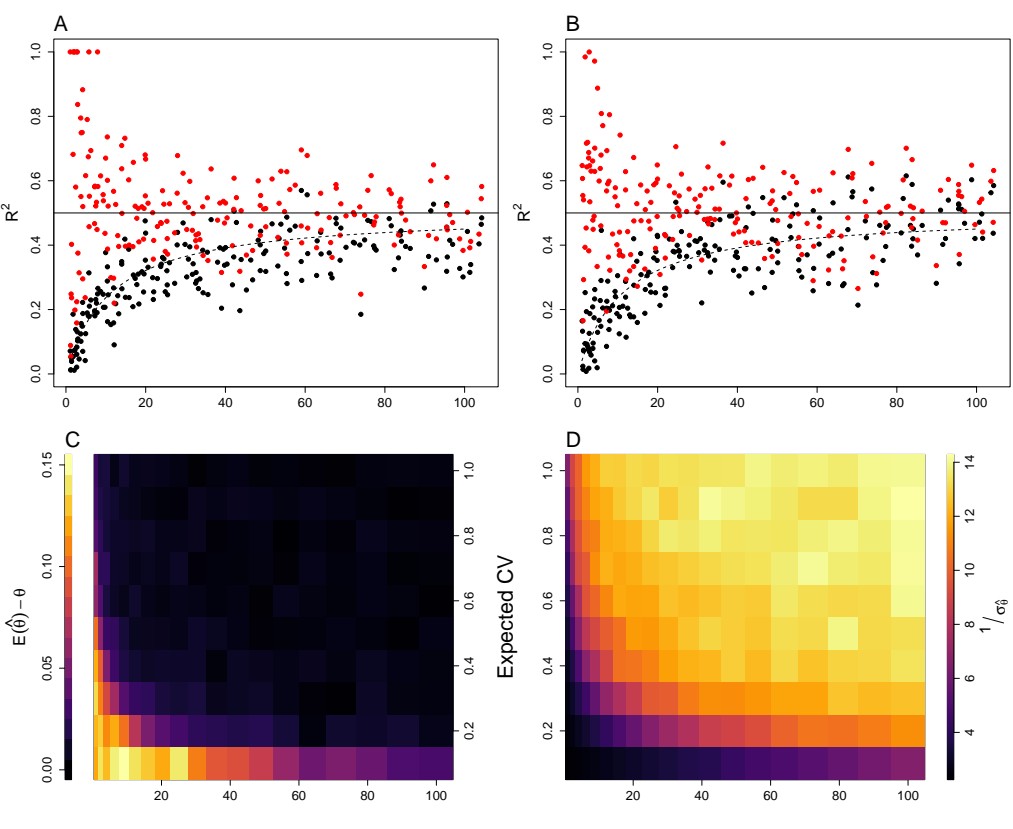

**Figure 6** **Results of simulations showing the effect of analysing behavioural count data with (red points) and without (black points) accounting for Poisson error, as both response (A) and predictor (B), over varying mean number of observed visits.** (A and B) were simulated with an expected CV of 0.3; solid lines show simulated $R^2$, and dotted lines show predicted $R^2$ when not taking in account Poisson error. (C and D) show bias and precision, respectively, in $R^2$ calculated from Poisson GLMM with provisioning rate as a response variable, from simulations across varying means and expected CVs; is the simulated value, and is the estimated value. Blue colours show low bias and low precision, respectively, and red colours high bias and high precision. See Supplementary Material S5 for further details of the simulations.

## Analyse number, rather than transforming to rate

In many studies the observation periods vary. This is frequently corrected for by scaling the observed number of events to create a rate (*e.g.*, visits/hour). In the provisioning literature, many studies also correct for brood size when calculating provisioning rate (*e.g.*, visits/hour/chick). Indeed, only 29% and 40% of studies in our literature survey (in 2015/16 and 2022 respectively) analysed their data as a count rather than a rate. However, when modelling count data, the raw number of observed events should be used, as transformations will create several problems. Firstly, because of the way the mean, stochastic error and expected variance scale(discussed above), by transforming the data the correct amount of Poisson variance cannot be directly estimated (Fig. 4). Once the number of observed arrivals is transformed to a different scale (*e.g.*, the number of

arrivals is standardised to arrivals/hour), the mean no longer represents the stochastic variance. For example, if the $20 \pm 7.5$ arrivals observed in study A from Fig. 4 (120 mins), is transformed to $10 \pm 3.75$ arrivals/hour, the expected CV (*i.e.,* the amount of biological variation) is calculated as 0.2 (instead of 0.3), as we are overestimating the amount of Poisson sampling error. Conversely, the $5 \pm 2.7$ arrivals in observations from study C (30 mins) when transformed becomes $10 \pm 5.4$ arrivals/hour, with an expected CV of 0.438. Therefore, depending on the direction of the scaling (*i.e.,* making the mean smaller or larger), this would lead to the respective over- or underestimation of Poisson variance, and so a corresponding over- or underestimation of effect sizes (see also Supplementary Materials S5 and S6). Instead, variation in observation time can be accounted for by using a Poisson 'exposure' model (*Gelman & Hill, 2007*), by including log observation period as an offset (a covariate with the slope fixed to 1). Similarly, brood size should be corrected for by including it as a covariate in the GLMM. Correcting provisioning rate for brood size (*e.g.,* using visits/hour/chick) incurs further problems associated with the use of ratios, as the relationship between brood size and provisioning rate is often not linear (*Raubenheimer, 1995*; *Nakagawa et al., 2017*), and spurious correlations are created when brood size is included as a covariate in addition to being corrected for in the response variable (*Kronmal, 1993*).

## Modelling interval lengths

Given some of the problems outlined above, it may seem more appealing to model the length of the intervals between behaviours rather than the count of the behaviours. Modelling interval lengths instead of counts is not a solution in itself, however. Clearly, there is an advantage to analysing interval lengths when auxiliary interval level data exists (such as environmental variables measured at the level of the interval), as this provides an opportunity to start to understand what processes contribute to the apparent stochastic nature of these interval lengths. However, auxiliary data are typically collected on the level of the observation and not the interval, which does not give more information to the analysis than analysing counts. Without interval-level data, the mean interval length for an observation is essentially the variable of interest, and this is clearly a simple re-parametrisation of the counts (mean interval length = observation period / count). As discussed above, we know that the interval lengths within an observation will show a high level of stochastic error (they are expected to be exponentially distributed under the Poisson process model). They can be treated as repeated measurements, where the within observation variation in interval lengths represents the stochastic error.

Given our extensive discussion above, we would encourage the use of a gamma distribution when modelling behavioural interval lengths. This approach also allows a population level $\alpha$ to be estimated, and so some of the assumptions of a Poisson process can be directly assessed. An interesting extension to this, would be to model whether both the refractory period and the return rate vary between observations and further whether they systematically differ between individuals or environments.

## DETERMINING SAMPLING EFFORT

Researchers frequently consider sample size when planning studies. In this case, that would typically refer to the total number of observations. However, as we have shown, the number of events that are observed within each observation is also important. How then should we determine the best strategy for collecting such data?

Readers may wonder why we do not simply recommend extremely short observation periods, given that we can correct for the additional stochastic error that is induced by this method. It is important to note, however, that when observing a low mean number of events, precision of model parameter estimates is low, and bias (under certain conditions) is high (Fig. 6). This is because when the mean number of observed visits is low, the estimation of the residual, unexplained biological variation is poor. As the variance is so dominated by stochastic error, small random fluctuations in the mean (induced by sampling error) have disproportionately large effects on the estimation of residual variance and can even lead to the mean being larger than the observed variance, implying that there is no variation in expected rates. We can see this pattern in both simulated data and in data from the literature survey (Supplementary Material S5 and Fig. S3). Moreover, this is likely why we see an upward bias in effect size at low mean values in the Poisson models, as the residual (*i.e.,* expected) variance is underestimated.

We should therefore seek to collect data under the conditions which minimise such effects. Ideally data would be collected in a fully automated way, meaning that all(or a large proportion of) events would be recorded, and the stochastic error across observations would be negligible. However, this is overly idealistic in most situations, as setting up such systems involves a large amount of time and money, and requires a high proportion of the population to be tagged to be effective. Thus, we advise using existing data (or a pilot study) to estimate a suitable observation period (see Supplementary Material S5 for how to calculate expected CV). This is not a one size fits all situation - an appropriate observation period will differ among study systems, according to the mean rate and variability of the behaviour. The emphasis, therefore, should be on the optimal mean number of observed events rather than optimal observation period, as it is the former that will directly determine the proportion of stochastic error. Our simulations show that an average of 20 events per observation minimises bias and maximises precision (but note that the results may vary according to parameters such as the simulated $R^2$). We recognise that longer observations may limit the number of observations that can be made, although researchers can use tools such as planned missing data designs (*Noble & Nakagawa, 2021*) to offset this cost.

Finally, it is worth noting that our calculations are made on the assumption that a Poisson process is the most suitable model for the behavioural count data in question, which may not be the case (see above). However, regardless of the exact form of the stochastic error, extending observation periods will act to reduce this error, and make comparisons between studies more meaningful.

## A CAUTIONARY NOTE ON OTHER MEASURES OF BEHAVIOUR

Whilst the literature survey presented here focused on studies that specifically measured and analysed counts of provisioning, there were many studies in our original search that analysed variables that are derived from visit rate, such as the amount of food brought to the nest or proportion of visits made by each sex. These metrics will equally be affected by the problems caused by stochastic error, as they depend on visit rate for their quantification, and therefore the stochastic error associated with visit rate is propagated to these other variables. We, therefore, would urge a similar word of warning in the use of any measures derived from short observations of behaviour. The kind of stochastic error we describe here applies not only to counts, but to any quantification of behaviour sampled in a short period of time. These problems can be resolved through careful thought about which distribution to use in the analysis, and the assumptions that a distribution has. For example, the amount of food brought to the nest might be well described by compound Poisson or Tweedie distributions (see *Thompson, 1984* for an example with rainfall data).

## CONCLUSIONS

1. Stochastic error arises when measuring behaviour by counting the frequency of events in a sample period. The degree of this error depends on both the number of events observed and the variation in rates between observations. By not taking this error into account, we limit both the variation in these behaviours that we can explain and the utility of these variables as predictors of other traits. Furthermore, as the degree of this error depends on characteristics of the study, comparisons between studies are highly problematic.

2. Using the null model of a Poisson process to describe this stochastic error, we can demonstrate it arises in a predictable manner, allowing researchers to account for it using established statistical methods. Whether, and how far, real behaviour count data deviates from this model is not well understood. Future work should seek to address this, as it will give a better biological understanding of the respective behaviour.

3. Using the example of provisioning rate, we demonstrate the suitability of the Poisson process as a null model. However, through a literature survey we show that by far the majority of studies of provisioning rate do not account for this stochastic error and, due to the low mean number of observations per study, the amount of stochastic error is high. Whilst recent work may be correct in asserting that provisioning rate is not an accurate descriptor of parental investment (*Williams, 2012*; *Williams & Fowler, 2015*), the methods that are currently employed to assess this are insufficient to draw this conclusion. Therefore, although we welcome further investigation of the different ways parents invest in their offspring (*e.g.*, size / quantity / quality of prey), we suggest that we should not yet rule out the possibility that provisioning rate itself, when properly measured, is an adequate description of postnatal parental investment. The use of longer observations and correct statistical analysis will aid us in these endeavours.

4. Finally, given the inevitability and predictable nature of this stochastic error, we should endeavour to quantify and account for it when analysing behavioural count data, as well as taking steps to minimise it where possible. Behavioural ecology is a discipline already fraught with relatively small effect sizes, and low power to detect them; we do ourselves a disservice by adding more error into the equation.

## ACKNOWLEDGEMENTS

We thank the LHB discussion group at the University of Sheffield for valuable input, and Daniel Noble, Fonti Kar, Alison Pick, David Westneat, Jarrod Hadfield, Paul Johnson and an anonymous reviewer for their comments on the manuscript.

### Funding

This work was supported by the Swiss National Science Foundation (P2ZHP3_164962 to Joel L. Pick), the Australian Research Council (FT130100268 to Shinichi Nakagawa) and the Natural Environment Research Council (MO 005941). The funders had no role in study design, data collection and analysis, decision to publish, or preparation of the manuscript.

### Grant Disclosures

The following grant information was disclosed by the authors:
The Swiss National Science Foundation: P2ZHP3_164962.
The Australian Research Council: FT130100268.
The Natural Environment Research Council: MO 005941.

### Competing Interests

The authors declare there are no competing interests.

### Author Contributions

- Joel L. Pick conceived and designed the experiments, performed the experiments, analyzed the data, prepared figures and/or tables, authored or reviewed drafts of the article, and approved the final draft.
- Nyil Khwaja performed the experiments, authored or reviewed drafts of the article, and approved the final draft.
- Michael A. Spence analyzed the data, authored or reviewed drafts of the article, and approved the final draft.
- Malika Ihle performed the experiments, authored or reviewed drafts of the article, and approved the final draft.
- Shinichi Nakagawa conceived and designed the experiments, authored or reviewed drafts of the article, and approved the final draft.

## Data Availability

All data and code for analyses and simulations presented here are available at Zenodo: Joel Pick. (2022). Data for: Counter culture: Causes, extent and solutions of systematic bias in the analysis of behavioural counts (v3.0). Zenodo. https://doi.org/10.5281/zenodo.7439115.

## Supplemental Information

Supplemental information for this article can be found online at http://dx.doi.org/10.7717/peerj.15059#supplemental-information.

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
