# Peer review of "Counter culture: causes, extent and solutions of systematic bias in the analysis of behavioural counts"

_PeerJ, doi:10.7717/peerj.15059_

## Round 0.1 · original submission · Major Revisions

I'd be willing to accept, albeit with significant revisions to bring the work up to date.

Reviewer 1 ·

Basic reporting

This manuscript documents the application of a Poisson model in the analysis of behavioural research. This is an interesting and useful study with some potential application across a broad range of study areas. I do have some concerns regarding the recency of the literature, as only three of the references were published within the '20s, and the vast majority of cited studies were published before 2017. I would therefore highlight this as an area for development.
The review is reasonably well structured, with a clear and logical progression to the points being made. The one area that needs development is the explanation of review methods (see experimental design).
The work is professional, with minimal evidence of grammatical errors (I have highlighted these on the PDF. Similarly, the supplementary document is instructive and useful (but the references would need revising as per stylistic requirements for PeeerJ.

Experimental design

As a review. There is limited discussion needed for experimental design. However, there are currently some issues with the explanation as to how literature was sourced. There needs to be evidence of keywords, sources and exact times for searching, and both inclusion and exclusion criteria are necessary. Without this information, it is not possible to repeat the literature search.
More concerningly, the literature search appears to be 2015-2016. There is no explanation at current as to why these dates were selected. It may be there is an excellent reason for these dates, but this, combined with the limited evidence of recent cited literature, might lead a reviewer to query as to when this paper was written. If this paper was produced five years ago, efforts need to be made to ensure the work is a reflection of current theory and practice in behavioural research.
This said, the research area is clear and the content has practical, meaningful applications. The figures that have been provided are clear to read and provide extra information for the manuscript.

Validity of the findings

The authors have admitted that the proposed model is only a simulation of the true, underlying mechanisms of behaviour, and they have made some well-considered arguments as to why this is necessary. There were some well-considered points regarding sample size issues and how they could be overcome; I think these would have enormous practical application to some researchers.

Additional comments

This is an interesting and useful study, but the currency of literature and arguments reduces the quality of the work. While the core elements of the Poisson arguments are sound, there needs to be substantial editing to ensure that this work is reflective of current practice and thinking in the field of behavioural counts. The authors should also ensure that the supplementary materials are formatted in line with the PeerJ author guidelines.

Annotated reviews are not available for download in order to protect the identity of reviewers who chose to remain anonymous.

·

Basic reporting

All good.

Experimental design

All good.

Validity of the findings

All good.

Additional comments

This is a useful, interesting and well written article on the importance of correctly modelling the sources of variation in behavioural count data when estimating statistics such as ICC and R-squared. The caveats below aside, it makes a convincing argument, backed up by a survey of the literature, and contains a wealth of good advice for modellers of count data.

Line 123
The justification of equation 1 needs to be more specific. It's stated that "given that variances (of independent variables) are additive" then var(y) is the sum of its components, the intrinsic Poisson variance s2stoc = lambda*t and the inter-observation (or inter-nest, using the authors' example) variance among the expected no of arrivals:

s2y = s2stoc + s2lt

The truth of this statement depends on the underlying model, which at this point hasn't been specified. If the model is additive, e.g. y_i ~ Pois(lambda_i) and lambda_i ~ N(0, s2lt) then equation 1 is true. However an additive model would be quite an unusual model for a Poisson RV, because (at least for most distributions) it's possible to produce negative lambda_i. A more common model would be to multiply lambda by e.g. gamma(theta, theta) (giving negative binomial) or by e^N(0, s^2) (giving Poisson-lognormal). In these cases the expression for s2y is quadratic (e.g. Elston et al. Parasitology 2001, 122, 563–569). Given that the authors' claim of bias rest on this argument, I think this section needs some added explanation, and the implications of this choice for the rest of the article should be explored. I'm pretty sure that the basic relationship of uncertainty due to stochastic error reducing with longer observation periods will hold true under either model (additive or multiplicative), but this needs to be checked and made clear to the reader. How does the shape of the relationship between no of arrivals and CV depend on this choice?

Fig 2
On a related point, I think it would help the reader match up the symbols and equations with their meanings if their values could be put on Fig 2A and B. E.g. in Fig 2A, I assume lambda = 10, t = 2, so s2stoc = 20, while s2lt = 0, whereas s2lt > 0 in 2B.

Line 139
The last term in equation one seems to have been re-named, without explanation, from sigma^2_lambda*t to sigma^2_exp. Please clarify/simplify.

Line 377
"Traditionally R2 and ICC are calculated on the observed scale (i.e. including stochastic error; Nakagawa and Schielzeth, 2010, 2013)."
I might have misunderstood what you mean by "observed scale", but I'm pretty sure that for Poisson GLMMs, the R-squared and ICC statistics defined by Nakagawa & Schielzeth are calculated on the (log) link scale, i.e. the scale where the variance components are additive, with only the distribution-specific error (here called stochastic error) variance having to be approximated via a transformation from the observed to the link scale.

Line 381
"In the case of variables such as provisioning rate (that are sampled in a fixed period of time), we would argue that these metrics should be estimated without the inclusion of this stochastic error in the estimation of total variance"
I wonder if it's worth making the point that these arguments apply equally to other distributions where the quantity of interest is a latent variable, e.g. logit- and probit-binomial models. I guess this is covered in the references, e.g. to Villemereuil et al., but it might be worth making it explicit.

Line 451
It would be useful to provide the reader with some introductory references to measurement error models in ecology. I don't know the literature very well but I'm aware that Stefanie Muff has published quite a bit on this (https://scholar.google.com/citations?user=KLm4qvYAAAAJ&hl=de), although there might be more suitable references out there.

Line 430
In addition to Tweedie, could a Conway-Maxwell-Poisson (http://doi.org/10.1890/13-1912.1) also be used to model the refractory period? Would it be worth mentioning this?

Fig 3 legend
The information on plot B is missing.

Line 257
Typo: Effects -> Effect

Throughout: R2 -> R^2

---

## Round 0.2 · Minor Revisions

Both reviewers have suggested some corrections. Kindly address them.

Reviewer 1 ·

Basic reporting

The paper is well written, with professional formatting. There are not problems with the grammar and text, however, I would have expected to see an update to the literature since the last review, given that the currency of references was highlighted as an area for development. This ultimately does not demonstrate that major updates in this field have been considered (or researched).
The content is well structured and figures are appropriate.

Experimental design

The authors have built upon the points raised on experimental design by factoring in a smaller, and more up-to-date, element to the study. This is useful in that it lends relevance to the older content. The relevance of the work is clear.

Validity of the findings

The findings remain valid as per the original submission of this paper. The methods are now more clearly described and this increases the repeatability of the work.

Additional comments

Dear Authors,
Thank you for submitting a revised copy of this paper. It is clear that the key comments arising from the original review have been addressed. The inclusion of the 29 papers pertaining to 2022 was good to see as it demonstrated similar findings to the older content. However, I was a little concerned to see that this update was not reflected in the manuscript's introduction or reference list. While it is promising to see the inclusion of this content into the analysis, this needs to be provided more clearly throughout the paper. The absence of current research means that any key changes in this field have been overlooked, and therefore an update is required here.

·

Basic reporting

All good

Experimental design

All good

Validity of the findings

All good

Additional comments

I'm happy with the authors' response all my points except my main one ("The justification of equation 1 ... depend on this choice?"), that the simple equation for the variance (equation 1) isn't sufficiently general to cover typical analyses of behavioural count data, and that this weakens their explanation of their point about diminishing stochastic error.

The authors responded:
"There is maybe some confusion here as to what lambda represents. At this point, we are not linking lambda to a statistical model. Lambda is the arrival rate and is bound to be positive. When statistically modelling this rate, we often assume that it has a lognormal or gamma distribution. However, we don’t specify the distribution of lambda at this point in the manuscript, as it is not relevant to this argument. Regardless of the distribution of lambda, the mean and variance in this distribution will change in a known way when multiplied by time, and the CV will remain constant, as shown in equation 2. Moreover, regardless of the distribution of lambda, the stochastic variance is always added to variance of lambda*t, as this is a general result regarding the sum of two independent random variables."

Although the authors say that "we are not linking lambda to a statistical model", it would be more accurate to say that the authors have partially specified a statistical model. Observed counts are Poisson, with mean and variance s2stoc = lambda*t. There is also variation in lambda*t among observations (nests), with this variance being labelled s2lt. It is implied (but should be clearly specified to avoid confusion) that this error is additive, in the sentence "First, given that variances (of independent variables) are additive, the total observed variance in the number of arrivals across observations (s2) is equal to the sum of this error (s2stoc) and the variance in expected number of arrivals (s2lt)
s2y = s2stoc + s2lt."
That the error is additive is implied because "variances (of independent variables) are additive" is true if the variables are summed (the authors should be more specific here by adding "sums of"), but other ways of combining RVs, such as multiplying, or defining a mixture of distributions (e.g. negative binomial or Poisson-lognormal) don't result in the variance being a simple sum of the component variances. Given that it is implied that this error is additive, this model must either allow negative expected nest counts (e.g. with Gaussian errors), or must have some unusual additive error distribution to avoid negative expected counts (for example, gamma errors, which are strictly positive, or some kind of truncated distribution dependent on lambda*t). None of the models implied is realistic, which undermines the overall point made by the authors. In their response, the authors write that "this is a general result regarding the sum of two independent random variables", but as I've pointed out, modelling counts with errors as the *sum* of two independent RVs (where one is Poisson) isn't sufficiently general to include any commonly used model of counts with errors.

Stepping back a bit... The authors are trying to explain what is an intuitive observation, that inter-nest (or group, cluster, etc) variation will not be diminished by collecting more within-nest data, but the amount of noise in the estimate of the within-nest arrival rates will diminish. If we were to specify a typical model of overdispersed counts, e.g. Poisson-lognormal or negative binomial, or even multiplicative, we'd find that equation 1 had a more complex form. For some distributions (e.g. negative binomial, Poisson-lognormal), the total variance would still have the form "lambda*t + ...", but lambda*t will also appear in the "..." part, so splitting the variances into two independent additive components isn't as simple as in equation 1. Other distributions (e.g. CMP) will have yet more complex variance functions. Therefore the example given by the authors isn't sufficiently general.

Although this undermines the explanation of the authors' overall point, I agree that the point they're making is valid. Typically, R2 and ICC for behavioural count data are estimated using variance components estimated on the log link scale via a GLMM, and if the stochastic variance is included, it has to be transformed approximately from the natural count scale to the link scale. The point the authors make about bias in R2 etc can be explained very clearly within this framework. Is it necessary to try to demonstrate greater generality? Could the authors explain their point within this log-link GLMM framework? Or could they make the more general point about diminishing within-group error relative to constant between-group error with a simple non-count model, before translating this point to a Poisson count model?

---

## Round 0.3 · accepted · Accept

My recommendation is to accept.

Reviewer 1 ·

Basic reporting

Improved since previous review. Some edits have been made to the abstract and some further citations have been added.

Experimental design

As per previous - no major edits are noted.

Validity of the findings

The findings are clear and well reported. No major changes have been made since the last review.